

# Investigating ultrafast quantum magnetism
# with machine learning

**Giammarco Fabiani** [★] **and Johan H. Mentink**

Radboud University, Institute for Molecules and Materials (IMM),
Heyendaalseweg 135, 6525 AJ Nijmegen, The Netherlands

★ gfabiani@science.ru.nl

## Abstract

We investigate the efficiency of the recently proposed Restricted Boltzmann Machine (RBM) representation of quantum many-body states to study both the static properties and quantum spin dynamics in the two-dimensional Heisenberg model on a square lattice. For static properties we find close agreement with numerically exact Quantum Monte Carlo results in the thermodynamical limit. For dynamics and small systems, we find excellent agreement with exact diagonalization, while for systems up to N=256 spins close consistency with interacting spin-wave theory is obtained. In all cases the accuracy converges fast with the number of network parameters, giving access to much bigger systems than feasible before. This suggests great potential to investigate the quantum many-body dynamics of large scale spin systems relevant for the description of magnetic materials strongly out of equilibrium.

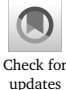

# 1 Introduction

Understanding the effect of correlations on the properties of quantum many-body systems is one of the most challenging problems of condensed matter physics today. In material science, the interest in this problem is rapidly growing, fueled by the availability of new advanced experimental techniques including ultrafast optical [1] and x-ray [2] spectroscopy. These methods allow to assess the dynamics of quantum spin correlations in magnetic materials, including transition metal oxides such as the parent compounds of the cuprates [3], as well as the transition-metal fluorides [4,5]. Clearly, theoretical methods to support and stimulate experiments on the dynamics of quantum spin correlations in magnetic materials are highly desired. However, even for the simplest relevant model system, i.e. the antiferromagnetic Heisenberg model on a square lattice, no analytical solutions are available.

Numerical methods generally offer a very powerful tool to get insights into quantum many-body systems and their dynamics. For example, at zero temperature, numerically exact results for both the ground state and dynamics can be obtained by using exact diagonalization (ED). However, with this approach the number of degrees of freedom scales exponentially with the system size, rendering it applicable only to systems with a small number of spins. This strongly limits the relevance to studying magnetic materials.

Starting from the one-dimensional limit, the exponentially large amount of information encoded in a quantum state can be efficiently compressed into a numerically tractable wavefunction. This is exploited in many successful algorithms, such as Density Matrix Renormalization Group [6], Matrix Product States [7] and more general Tensor Network States (TNS) [8]. On the other hand, Dynamical Mean Field Theory has proven to efficiently capture temporal correlations in high dimensional models and provide numerically exact results in the limit of infinite dimensions. However, in the intermediate cases of two and three dimensional systems, where both spatial and temporal quantum correlations are important, established methods are computationally demanding [9] and have limited capacity to simulate the time-evolution of non-local quantum spin correlations [10,11].

Recently a new wavefunction based method inspired by machine learning was proposed [12]. Here the quantum many-body states are represented by means of a Restricted Boltzmann Machine (RBM), which is a generative and stochastic Artificial Neural Network (ANN) featuring one input and one hidden layer. Intriguingly, this method can be applied to efficiently simulate temporal and spatial correlations in any dimension even in the case of highly correlated states [13], where generally TNS-based algorithms become inefficient. This suggests great potential for the study of strongly non-equilibrium dynamics in models relevant to strongly correlated systems in two and three dimensions. However, the RBM ansatz has been applied to simulate dynamics only in one-dimensional systems, therefore it is not clear how accurate and efficient it is in higher dimensions.

In this work, we apply the RBM ansatz to study static properties and simulate the dynamics of the antiferromagnetic Heisenberg model on a square lattice. First, by optimizing the neural network parameters in the static case we confirm the results obtained before [12] and extend this to larger system sizes, which allows us to extrapolate to the thermodynamical limit where we find close correspondence with numerically exact quantum Monte Carlo. Second, for dynamics we show that the RBM ansatz can simulate non-trivial unitary dynamics for long evolution times and for system sizes well beyond ED. To validate these findings, we compare the results with analytical calculations based on the Random Phase Approximation, finding close agreement. Finally, we estimate the system sizes that are accessible with this method. In the appendix, we provide a self-contained description of the algorithm and our implementation in Julia. An open source version of this code termed "ULTRAFAST" is provided in [14].

## 2 The Restricted Boltzmann Machine representation

In this section we introduce the Restricted Boltzmann Machine (RBM) representation and outline how it is trained to describe quantum correlations efficiently. The RBM representation is formed by supplementing the physical system of spins $S_i$ with an auxiliary layer of Ising spins $h_j$. Each Ising spin is connected to all physical spins by parameters $W_{ij}$ to describe correlations between the spins in the physical layer. The probability amplitude to observe a particular spin configuration $S = (S_1, \ldots, S_N)$ in such a network is given by

$$\mathcal{P}(S) = \sum_{\{h_j\}} e^{\sum_{i=1}^{N} a_i S_i^z + \sum_{j=1}^{M} b_j h_j + \sum_{ij} W_{ij} h_j S_i^z}, \tag{1}$$

where the sum over $\{h_j\}$ means a trace over all the auxiliary spins. In the language of artificial neural networks, $S_i$ and $h_i$ are the visible and hidden units, respectively; the set $W_{ij}$ are artificial synapses, $a_i$ ($b_i$) are the visible (hidden) biases, and $M$ denotes the number of hidden units. Following [12], the wavefunction of the quantum spin system is identified with the probability amplitude Eq. (1), namely $\langle S|\psi_M \rangle \equiv \psi_M(S) = \mathcal{P}(S)$, and the network parameters are extended to complex values to allow $\psi_M(S)$ to represent negative (or complex) probability amplitudes. A RBM has no intra-layer connections and therefore the hidden degrees of freedom can be easily traced out, obtaining for the wavefunction the following ansatz

$$\psi_M(S) = e^{\sum_{i=1}^{N} a_i S_i^z} \times \prod_{i=1}^{M} 2\cosh\left(b_i + \sum_j W_{ij} S_j^z\right). \tag{2}$$

Since the exact wavefunction is in general unknown, the set of network parameters $\mathcal{W}_k = \{a_i, b_i, W_{ij}\}$ is trained via a variational Monte Carlo algorithm: at each step, spin states are sampled from the Hilbert space and used by the network to generate feedback based on variational principles. The optimization criterion can be derived in different ways. For dynamics we adopt a time dependent variational scheme where at each time-step the Hilbert space distance $\mathcal{R}(\mathcal{W}(t)) = \text{dist}\left(\partial_t |\psi_M(t)\rangle, -i\hat{H}|\psi_M(t)\rangle\right)$ is minimized. This leads to a set of ordinary differential equations for the network parameters

$$S_{kk'}(t)\dot{\mathcal{W}}_{k'}(t) = -i\mathcal{F}_k(t), \tag{3}$$

where $S_{kk'}$ and $\mathcal{F}_k$ are defined in terms of the derivatives of the RBM wavefunction with respect to $\mathcal{W}_k$ [15]. Ground state optimization is obtained similarly, by replacing real time with imaginary time and the optimization routine becomes equivalent to a norm-independent minimization of the expectation value of the energy. Further details are given in the Appendices A and B.

The computational cost of the RBM approach is determined by the dimension of the variational manifold: $N_{var} = N + M + M \times N$, where $M = \alpha N$; the integer $\alpha$ will set the capacity (and accuracy) of the network. By exploiting symmetries of the Hamiltonian it is possible to lower the dimension of the variational manifold. For instance, in the case of full site-translation symmetry that we exploit below, the number of independent parameters reduces to $N_{var} = 1 + \alpha + \alpha N$.

## 3 Ground state calculations

In this work we consider the antiferromagnetic Heisenberg model on a square lattice with $N = L \times L$ physical spins $\hat{S}_i = \hat{S}(\mathbf{r}_i)$, with $\mathbf{r}_i = (x_i, y_i)$

$$\hat{H} = J_{ex} \sum_{<ij>} \hat{S}_i \cdot \hat{S}_j, \tag{4}$$

where $J_{ex}$ is the exchange interaction ($J_{ex} > 0$) and $\langle \cdot \rangle$ restricts the sum to nearest neighbours. Periodic boundary conditions are employed in what follows. Since the model under study is bipartite, we can perform a gauge transformation corresponding to a rotation of one sublattice, which changes the sign of the off-diagonal terms of Eq. (4). This transformation results into positive probability amplitudes $\langle \psi_M | s \rangle$ in the ground state and allows to use real-valued variational parameters for the optimization of $\psi_M$. Therefore for static calculations we will adopt real-valued weights and biases. Below we present results for the ground state energy per spin and the staggered magnetization defined as

$$E(L) \;=\; \frac{1}{L^2} \frac{\langle \psi_M | \hat{H} | \psi_M \rangle}{\langle \psi_M | \psi_M \rangle}, \tag{5}$$

$$M(L) \;=\; \frac{1}{L^2} \sum_{j=1} (-1)^{\|\vec{r}_j\|} \hat{S}_j^z, \tag{6}$$

respectively, where the dependence on $L$ is also implicit in $\hat{H}$ and $|\psi_M\rangle$.

We are interested in the behaviour of $E(L)$ and $M(L)$ for $L \rightarrow \infty$, which we extrapolate from finite size scaling [16]. For the energy we have

$$E(L) = E(\infty) + a L^{-3} + \dots. \tag{7}$$

The extrapolation of the staggered magnetization is more subtle since $M(L)$ is zero in a finite lattice because the model is isotropic. Therefore, we estimate $M(\infty)$ from the spin correlation functions $\langle \hat{S}_i \cdot \hat{S}_j \rangle (L) = \langle \psi_M | \hat{S}_i \cdot \hat{S}_j | \psi_M \rangle$ (the $L$-dependence is again in $\psi_M$) using that $|\langle \hat{S}_i \cdot \hat{S}_j \rangle| - M^2 \sim 1/r_{ij}$ in the limit of large $r_{ij} = \| \vec{r}_i - \vec{r}_j \|$ [17]. If we choose $\hat{S}_j = \hat{S}_{i+R} \equiv \hat{S}(\vec{r}_i + \vec{R}_L)$, with $\vec{R}_L = \left(\frac{L}{2}, \frac{L}{2}\right)$, then in the thermodynamical limit $r_{ij} \longrightarrow \infty$ and we can identify $M^2(\infty)$ with $|\langle \hat{S}_i \cdot \hat{S}_j \rangle|(\infty)$. The value of the spin correlation at infinity can be extrapolated using the following scaling behaviour [16]

$$|\langle \hat{S}_i \cdot \hat{S}_{i+R} \rangle|(L) = |\langle \hat{S}_i \cdot \hat{S}_{i+R} \rangle|(\infty) + c L^{-1} + \dots. \tag{8}$$

In both Eq. (7) and Eq. (8), we retain only the leading order correction.

Fig. 1(a) shows the scaling of the energy with the system size and density of hidden units $\alpha$. As expected $E(L)$ decreases with increasing $\alpha$. The relative error in the energy $\epsilon_{rel} = (E_{RBM} - E_{QMC})/|E_{QMC}|$ as a function of $\alpha$ is plotted in Fig. 1(b); the QMC result $E_{QMC} = -0.669437(5)$ is used as a reference [16]. Similar as demonstrated before for $N = 100$ [12], for $N = 144$ the RBM representation outperforms one of the most accurate variational methods (PEPS [18], horizontal dashed line) already for a modest number of hidden units. Fig. 1(c) shows similar convergence with alpha for the energy $E(L)$ in the limit $L \longrightarrow \infty$, as extrapolated from the fits shown in Fig. 1(a).

The extrapolation of $M$ for large $L$ is plotted in Fig. 1(d) together with results obtained from spin wave theory (SWT) and QMC [16,17]. The finite correlation functions are calculated employing $\alpha$ in a range of values between 8 and 16 until convergence was achieved. Numerical data for $E(L)$ and $\langle \hat{S}_i \cdot \hat{S}_{i+R} \rangle (L)$ are provided in Appendix D.

## 4 Spin dynamics

In this section we study the efficiency of the RBM ansatz for the description of the spin dynamics of Heisenberg antiferromagnets. For small system size ($N = 16$), we compare the RBM results with ED results obtained using QuSpin [19] and for larger systems we compare it with interacting spin-wave theory. In particular, we focus on Raman scattering of pairs of

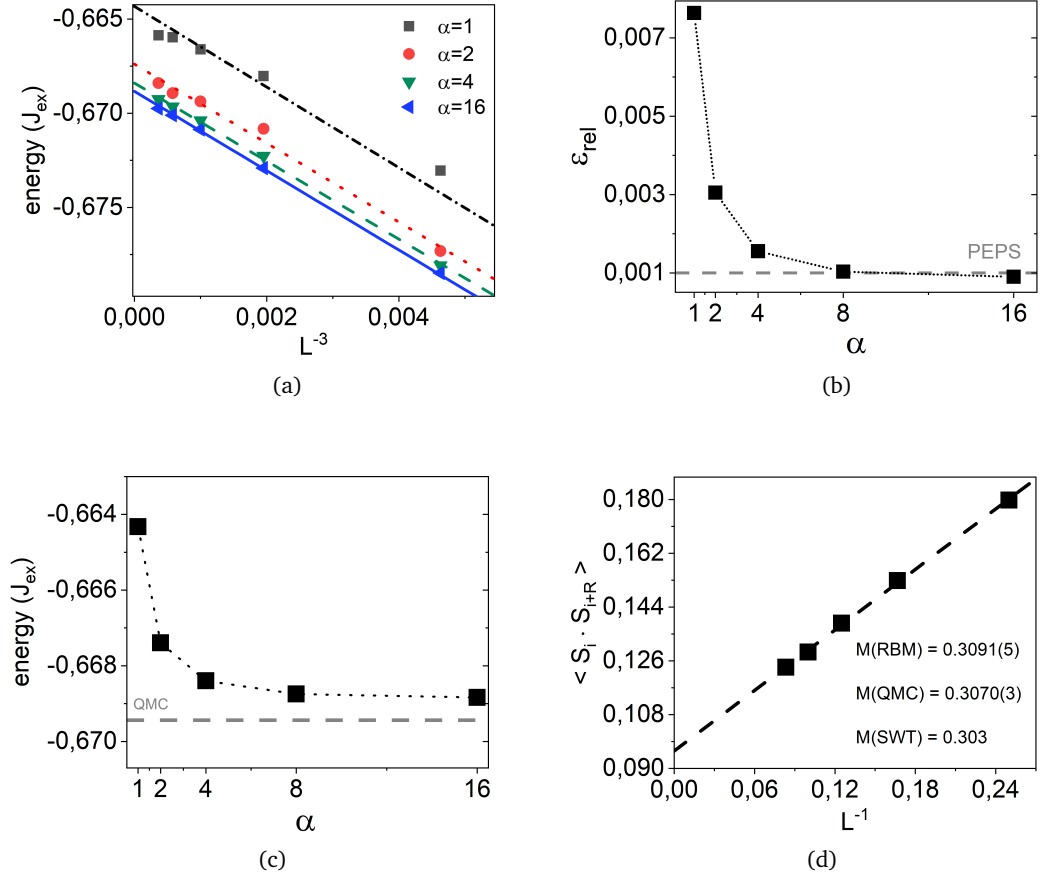

Figure 1: (a) Scaling of the ground state energy with system size $L$ and density of hidden units $\alpha$. The lines are linear fits through the numerically obtained data points. (b) Relative error $\epsilon_{rel}$ with respect to QMC as function of $\alpha$ for $N = 144$; the dashed line shows the error for PEPS [18]. The extrapolations from the fits for several $\alpha$ are shown in (c) and compared with QMC (dashed gray line) (d) Scaling of the spin correlation between the furthest spins in the lattice with system size. The dashed line is a linear fit from which the staggered magnetization $M$ is obtained. The extracted $M$ is shown in the inset and is found to be very close to the numerically exact QMC result [16]. For comparison also $M$ obtained from linear spin-wave theory is given.

spin excitations, the so-called two-magnon modes. For the simple cubic lattice we use the following time-dependent perturbation of the spin Hamiltonian (i.e. the Raman scattering operator) [20–23]

$$\delta\hat{\mathcal{H}} = \Delta J_{ex}(t) \sum_{i,\boldsymbol{\delta}} \left(\mathbf{e} \cdot \boldsymbol{\delta}\right) \hat{S}(\mathbf{r}_i) \cdot \hat{S}(\mathbf{r}_i + \boldsymbol{\delta}), \qquad (9)$$

where $\mathbf{e}$ is a unit vector that determines the orientation of the electric field which causes the perturbation and $\boldsymbol{\delta}$ connects nearest neighbour spins.

Our main interest is the study of impulsively stimulated Raman scattering that was recently investigated both experimentally and theoretically on the basis of harmonic magnon theory [4, 5, 22]. To model this problem, we approximate the time-dependent change of the exchange interaction as a square pulse with height $\Delta J_{ex}$ and temporal width $\tau$. We use $\Delta J_{ex} = (0.05 \div 0.1) J_{ex}$ and $\tau = 0.2/J_{ex}$ and we set $\mathbf{e}$ along the $y$-direction of the lattice. Simulations always start from the variational ground state obtained at the given $J_{ex}$. The

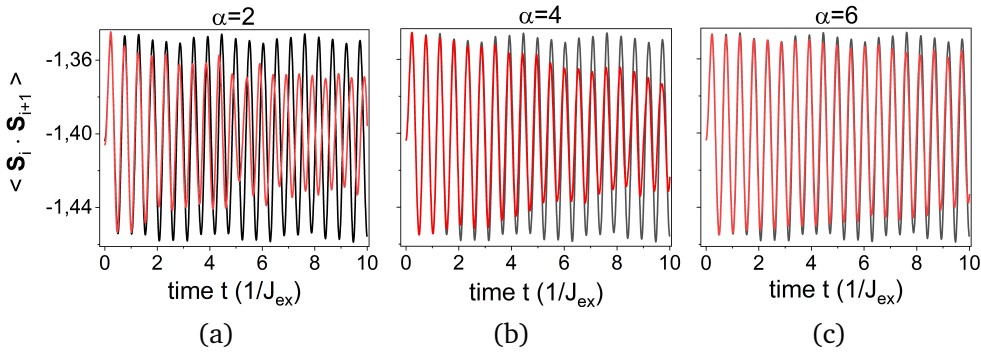

Figure 2: Spin-spin correlation functions between nearest neighbours spins in a $4 \times 4$ system for different $\alpha$ along the direction normal to the direction of perturbation Eq. (9). The red line is the RBM result, while the black line the ED result. At times $t \lesssim 2$ the RBM dynamics shows good overlap with the dynamics from ED even with $\alpha = 2$. For large simulation times the overlap rapidly improves with $\alpha$.

algorithm provides (complex valued) time-dependent weights that are subsequently used to evaluate observables $\langle \hat{O}(t) \rangle = \langle \psi_M(t) | \hat{O} | \psi_M(t) \rangle$ via Monte Carlo sampling. We note that the perturbation term Eq. (9) does not break the translation invariance and therefore translation symmetry is employed in the time-dependent RBM wavefunction. For observables, we evaluate spin-spin correlation functions $\langle \hat{S}_i(t) \cdot \hat{S}_j(t) \rangle$ which evolve non-trivially after the perturbation. Motivated by time and frequency resolved Raman scattering experiments we also evaluate the spin structure factor

$$S(\mathbf{q}, t) = \frac{1}{N} \sum_{ij} e^{i\mathbf{q} \cdot (\mathbf{r}_i - \mathbf{r}_j)} \langle \hat{S}_i(t) \cdot \hat{S}_j(t) \rangle, \tag{10}$$

which is closely related to experimental techniques such as resonant inelastic x-ray scattering [2,3,24–26]. Here the sum extends over all the possible pairs and $\mathbf{q}$ is a vector in the reciprocal space of the lattice. Moving to the frequency domain we evaluate the $\mathbf{q}$-integrated structure factor

$$\sum_{\mathbf{q}} S(\mathbf{q}, \omega) = \sum_{\mathbf{q}} \int dt \, e^{i\omega t} S(\mathbf{q}, t), \tag{11}$$

which filters the frequencies of all the modes excited and can be directly compared with interacting spin-wave theory and optical Raman spectra [27].

First, results for the $4 \times 4$ system are presented. Fig. 2 plots the time evolution of nearest neighbour spin correlations for different $\alpha$ and with $\Delta J_{ex} = 0.05 J_{ex}$. As it can be seen from the figure, already with $\alpha = 2$ the RBM representation matches well the exact result for $t \lesssim 2$ and the accuracy at later times rapidly improves with $\alpha$. The artificial damping with time appearing in Fig. 2 can have several sources. A Monte Carlo error due to limited sampling of spin configurations has been ruled out using all the states of the Hilbert space, which is still feasible in a $4 \times 4$ system; also with full sampling the dynamics obtained with the RBM ansatz exhibits this dissipation. Possible errors originating from the numerical time integration of Eq. (3) have also been ruled out by checking different integration schemes and systematically decreasing the step-size until convergence was achieved. Another possible error can originate from the iterative solver used to invert the matrix $S$ in Eq. (3) which in general can be singular. However, the same dissipative behaviour was observed for different inversion schemes or by regularizing the singularities as done in ground state optimizations (see Appendix A). Moreover, a dependence on the inversion scheme is not expected to improve by increasing the number of hidden units. Therefore, we believe that the dominant source of the artificial damping is in the representability of the RBM ansatz: at finite $\alpha$, there is a finite error of the

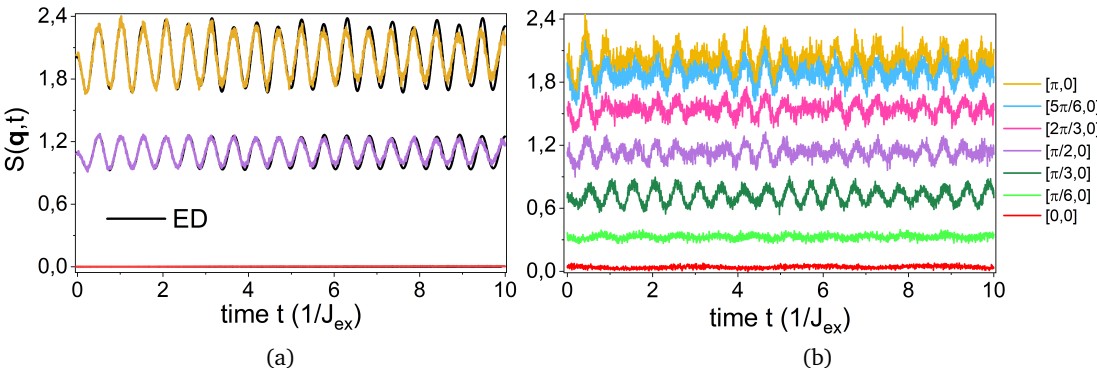

Figure 3: (a) Dynamics of the structure factor for $L = 4$, $\alpha = 10$. Excellent agreement with ED (solid black line) is found. (b) Similar dynamics for $L = 12$, $\alpha = 8$. In both cases all the non-equivalent modes $\mathbf{q} = [q, 0]$ in the first Brillouin zone are plotted. Note that the number of such modes depends on the system size. The noise in (b) is due to a limited Monte Carlo sampling.

time-evolved RBM wavefunction, which propagates during the time evolution and can cause the discrepancies observed. This error can be reduced by increasing the expressive power of the RBM representation, which is indeed confirmed by the numerical results.

Similarly, good convergence is achieved for other spin correlation functions and slightly larger perturbations. This is shown in Fig. 3(a) where the structure factor for $\Delta J_{ex} = 0.1 J_{ex}$ is plotted. In this case, $\alpha = 10$ was needed to achieve good correspondence with ED. The results in Fig. 3(a) prove that the RBM ansatz is able to catch not only the correlations between nearest neighbours, but also all the other correlations in the system and their time evolution.

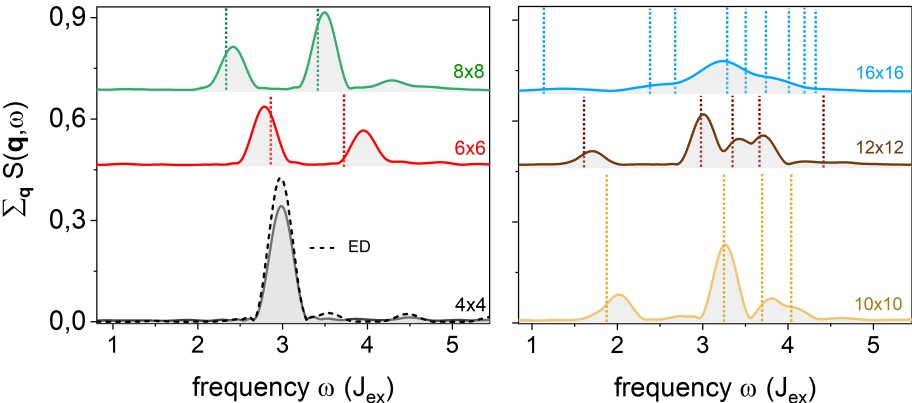

Figure 4: Integrated structure factor for different system sizes and $\Delta J_{ex} = 0.1 J_{ex}$ (solid lines). Data for $4 \times 4$ are compared with ED (dashed black line) showing excellent agreement in the position of the main peak. Data for larger sizes are compared with the frequency peaks of the excited modes obtained with interacting spin wave theory (dotted vertical lines). The Fourier transform to the frequency domain exploits a time window with total time length $t_{max} = 10/J_{ex}$ for system sizes up to $12 \times 12$; for $16 \times 16$, $t_{max} = 5/J_{ex}$. The RBM ansatz captures all the modes from RPA and closely resembles their position in the frequency domain.

Next, we study the dynamics for larger systems up to N=256, with $\Delta J_{ex} = 0.1 J_{ex}$. For these

system sizes, the same perturbation generates oscillations of the spin correlations which have smaller amplitude than the N=16 case previously examined. Therefore the simulation of larger systems becomes an easier problem for the RBM ansatz and already at $\alpha = 4$ convergence is reached within the Monte Carlo error. Fig. 3(b) shows the dynamics of $S(\mathbf{q}, t)$ for all non-equivalent modes $\mathbf{q} = (q, 0)$ in the first Brillouin zone of the $12 \times 12$ system.

Fig. 4 shows the integrated structure factor for different system sizes together with results from RPA calculations [27]. It is well known from interacting spin-wave theory in the thermo-dynamical limit, that the structure factor is a continuum of modes that peaks slightly below $\omega_R = 4J_{ex}$ due to a van Hove singularity at the Brillouin zone boundary. Hence, the dominant contribution of this peak originates from modes with large wave numbers that can be well captured in finite systems. For finite system size, it is therefore expected that the structure factor shows a few peaks around $\omega_R$. This is indeed observed in Fig. 4. For $N = 16$, again excellent agreement with ED is obtained, in particular for the position of the main peak. For larger systems, we observe that all the modes obtained from the RPA calculation [27] are present as well in the RBM results. The shifts in the position of the peaks with respect to RPA can be ascribed either to an inaccuracy of the RBM results or to an intrinsic error of the RPA method (or to a combination of both). The width of the peaks is due to the finite total integration time.

## 5   Conclusions

In this paper we have assessed both the ground state and dynamics of the 2D Heisenberg model. By comparison with numerically exact results, rapid convergence with neural network parameters is found. Moreover, for dynamics the RBM ansatz is able to capture all the magnon modes found from interacting spin-wave theory with a modest number of hidden units. This proves that it can efficiently simulate the protocol under study.

The current results show that systems up to $N = 256$ spins are feasible, and our implementation can efficiently simulate even larger systems as well. This is due to the fact that for fixed $\alpha$, the CPU time of the optimization scales only quadratically with the system size $N$ (see Appendix D) and therefore it can be contained exploiting parallelization. Moreover, CPU time can be reduced further exploiting other symmetries of the system on top of the translation invariance. In particular we checked that for $\alpha \leq 10$ and $\alpha N \leq 10^4$ [28], system sizes up to $30 \times 30$ spins are feasible in reasonably accessible CPU time on our local cluster nodes. Such system sizes are far beyond the capabilities of exact diagonalization.

Beyond the RMB ansatz studied here, it would be interesting to benchmark against more advanced neural quantum states [29–32]. Moreover, it will be very interesting for future applications to study more realistic spin models, including additional exchange interactions [33], which also requires different geometries of the perturbation operator.

While for the present work the dynamics simulations are focused on the linear response regime, the rapid convergence with $\alpha$ suggests the possibility to study spin correlations in antiferromagnets strongly out of equilibrium, where other state-of-the-art methods are severely limited. This also suggests that the RBM ansatz has great potential for disclosing novel phenomena based on the ultrafast quantum dynamics of antiferromagnets.

## Acknowledgements

**Funding information**   This work received funding from the Nederlandse Organisatie voor Wetenschappelijk Onderzoek (NWO) by a VENI grant and is part of the Shell-NWO/FOM-initiative "Computational sciences for energy research" of Shell and Chemical Sciences, Earth

and Life Sciences, Physical Sciences, FOM and STW.

# Appendices

## A   Details on the algorithm

In this Appendix we provide a self-contained description of the machine learning algorithm used in the main text. Using the same notation, a generic quantum state of a spin system can be efficiently parametrized by the RBM ansatz

$$\psi_M(S) = e^{\sum_i a_i S_i^z} \times \prod_{i=1}^{M} 2\cosh\left(b_i + \sum_j W_{ij} S_j^z\right). \tag{A.1}$$

$a_i$, $b_i$ and $W_{ij}$ are a set of respectively $N$, $\alpha N$, $M \times N$ parameters, where $\alpha = M/N$ is an integer representing the density of hidden units [12]. The RBM wavefunction can be interpreted as a black box, which receives a configuration state $|S\rangle$ and outputs the projection of the wavefunction onto this state. The output is determined by the network parameters and they have to be optimized according to the physical state we want the wavefunction to describe. In our simulations the input spin configuration is an array of $N$ elements, each of them taking the value $+1$ or $-1$.

The optimal choice for the set $\mathcal{W}$ is given by a reinforcement learning algorithm supplied with a Markov chain Monte Carlo sampling, which in turn is nothing else than a variational Monte Carlo approach. For a fixed set $\mathcal{W}$, the expectation value of an observable $\hat{A}$ is given by

$$\langle \hat{A} \rangle = \frac{\langle \psi_M | \hat{A} | \psi_M \rangle}{\langle \psi_M | \psi_M \rangle} = \frac{\sum_S |\psi_M(S)|^2 A_{loc}(S)}{\sum_S |\psi_M(S)|^2}, \tag{A.2}$$

where

$$A_{loc} = \frac{\langle S | \hat{A} | \psi_M \rangle}{\langle S | \psi_M \rangle}. \tag{A.3}$$

If we interpret the quantity $P(S) \equiv |\psi_M(S)|^2 / \sum_S |\psi_M(S)|^2$ as a probability density (note that $P(S) \geq 0$ and $\sum_S P(S) = 1$), we can engineer a Markov chain which has $P(S)$ as equilibrium distribution. In particular, starting from a (random) spin configuration $|S\rangle$, a chain of states can be generated with a Metropolis-Hastings algorithm where at each step one or two spins are flipped according to the acceptance

$$C\left(S_k \longrightarrow S_{k+1}\right) = \min\left(1, \frac{|\psi_M(S_{k+1})|^2}{|\psi_M(S_k)|^2}\right). \tag{A.4}$$

The excitation protocol Eq. (9) does not mix different magnetization sectors. Therefore since in the ground state the total magnetization is zero, we input an initial spin configuration with zero net magnetization and we look for spin-flips of two opposite spins in such a way that the magnetization is kept fixed.

After a certain equilibration time, configuration states are sampled according to the distribution $P(\{S\})$ and quantum expectation values can be estimated as

$$\langle \hat{A} \rangle \simeq \frac{1}{N_s} \sum_{k=1}^{N_s} A_{loc}(S_k), \tag{A.5}$$

where $N_s$ is the number of states visited by the Markov chain sampling and it is arbitrarily chosen according to the system size and $\alpha$. We generally follow the rule of thumb $N_s > 10\, N_{var}$ [28].

So far we have not described how to obtain the optimized variational parameters $\mathcal{W}$. This is done through a minimization procedure which generally differs for ground state and unitary dynamics, although at the end we will show that the two algorithms are closely related.

The ground state wavefunction of a given spin Hamiltonian is found via the Stochastic Reconfiguration method introduced in [34] and applied to the RBM wavefunction in [12]. This is a variant of gradient descent-like methods where at each step $p$ the variational parameters are updated according to the rule

$$\mathcal{W}_k(p+1) = \mathcal{W}_k(p) - \gamma(p) S_{kk'}^{-1} \cdot \mathcal{F}_k(\mathcal{W}(p)), \tag{A.6}$$

with

$$S_{kk'} = \langle O_k^* O_{k'} \rangle - \langle O_k^* \rangle \langle O_{k'} \rangle, \tag{A.7}$$

$$\mathcal{F}_k = \langle E_{loc} O_k^* \rangle - \langle E_{loc} \rangle \langle O_k^* \rangle. \tag{A.8}$$

Here

$$O_k(S) = \frac{1}{\psi_M(S)} \partial_{\mathcal{W}_k} \psi_M(S), \tag{A.9}$$

$$E_{loc}(S) = \frac{\langle S | \hat{H} | \psi_M \rangle}{\psi_M(S)}. \tag{A.10}$$

The (Hermitian) covariance matrix $S_{kk'}$ is in general non-invertible and therefore $S_{kk'}^{-1}$ strictly denotes the Moore-Penrose pseudo-inverse. To stabilize the inversion of the $S$-matrix we adopt the following regularization: $S_{kk} \rightarrow S_{kk} + \epsilon$, with $\epsilon \sim 10^{-4}$ and a constant step-size $\gamma(p) \sim 10^{-3}$. More advanced regularizations or choices for $\gamma(p)$ are possible [12], but we found our choice stable and efficient within our model.

The evaluation of the quantum expectation values are done with the Monte Carlo procedure outlined above. Usually a few hundreds of steps are needed to converge towards the ground state. Convergence is monitored by measuring the variance of the energy $\sigma_E^2 = \langle \hat{H}^2 \rangle - \langle \hat{H} \rangle^2$, which vanishes in the exact ground state. In practice, it is not guaranteed that the RBM ansatz convergences to the exact solution and the RMB solution is considered converged when the energy variance does not decrease further below the Monte Carlo error.

Unitary dynamics follows from the Time-Dependent Variational Principle (TDVP) applied to the RBM wavefunction. It is based on the minimization with respect to the variational parameters of the residual distance

$$\mathcal{R}(\mathcal{W}(t)) = \text{dist}(\partial_t |\psi_M\rangle, -i\hat{H} |\psi_M\rangle). \tag{A.11}$$

In Appendix B we show that this yields a set of ordinary differential equations for the variational parameters

$$S_{kk'}(t) \dot{\mathcal{W}}_{k'}(t) = -i \mathcal{F}_k(\mathcal{W}(t)), \tag{A.12}$$

where $S_{kk'}(t) = S_{kk'}(\mathcal{W}(t))$. To solve Eq. (A.12) we adopt a second-order time integration scheme based on the Heun scheme

$$\widetilde{\mathcal{W}}(t + \delta t) = \mathcal{W}(t) - i\delta t \, S^{-1}(t) \mathcal{F}(\mathcal{W}(t)) \tag{A.13}$$

$$\mathcal{W}(t + \delta t) = \mathcal{W}(t) - \frac{i\delta t}{2} \left[ S^{-1}(t) \mathcal{F}(\mathcal{W}(t)) + \widetilde{S}^{-1}(t + \delta t) \mathcal{F}(\widetilde{\mathcal{W}}(t + \delta t)) \right], \tag{A.14}$$

where $S^{-1}$ is obtained from Eq. (A.12) using the iterative solver MINRES [35], which is found to be stable throughout the whole dynamics and $\widetilde{S}(t + \delta t) = S(\widetilde{\mathcal{W}}(t + \delta t))$.

We generally choose $\delta t$ in the range $[0.0025, 0.005]/J_{ex}$. No improvements have been observed in the dynamics when using a smaller time-step. Higher orders integration schemes

have been investigated (for instance 4th Runge-Kutta) but no further improvements on the efficiency and accuracy respect to the Heun scheme have been observed.

Since the energy is a conserved quantity after the double quench, we keep track of the quality of the simulation looking at the the time-evolution of the energy. Large jumps in the energy signal breakdowns in the simulation, or large deviations from the initial value can result in a large loss of accuracy in the time-evolution.

To conclude this Appendix we note that the ground state optimization rule is equivalent to the time-dependent variational principle (Eq. (A.12)) applied in imaginary time and solved with an Euler integration scheme. This means that the SR method solves for the time-evolution induced by $U = e^{-\tau \hat{H}}$ which always converges for large $\tau$. In this approach Eq. (A.6) gives at each step the parameters of the imaginary time-evolved wavefunction $|\psi_M(\tau + \delta\tau)\rangle = e^{-\delta\tau\hat{H}}|\psi_M(\tau)\rangle$.

# B  The time-dependent variational principle

In this Appendix we show that the time-dependent variational principle Eq. (A.12) can be derived both from minimization of the residual distance and from a Lagrangian formulation. In the former case we start from the residual distance which we write down explicitly

$$\mathcal{R}\big(\mathcal{W}(t)\big)^2 = \left\lVert \Big(1 - \frac{|\psi_M\rangle\langle\psi_M|}{\langle\psi_M|\psi_M\rangle}\Big)\Big(i\frac{d}{dt}|\psi_M\rangle - \hat{H}|\psi_M\rangle\Big)\right\rVert^2, \tag{B.1}$$

where $\lVert \cdot \rVert$ indicates the norm in the Hilbert space where the RBM wavefunction is defined. The second term in the first parenthesis in the right hand side enforces conservation of the norm in the minimization, leading to a norm-independent dynamics. Working out the expression we find that

$$\mathcal{R}\big(\mathcal{W}(t)\big)^2 = \dot{\mathcal{W}}_k^* \dot{\mathcal{W}}_{k'} S_{kk'} - i\dot{\mathcal{W}}_k \mathcal{F}_k^* + i\dot{\mathcal{W}}_k^* \mathcal{F}_k + \langle\hat{H}^2\rangle - E^2. \tag{B.2}$$

Minimizing with respect to $\dot{\mathcal{W}}_k^*$ yields the TDVP equations of motion (3). We note that $\mathcal{R}\big(\mathcal{W}(t)\big)$ is related with the Fubini-Study metric introduced in [12] by $\mathcal{R}_{FS} \equiv \text{dist}_{FS}\big(\mathcal{W}(t)\big) = \arccos\sqrt{1 - \delta t^2 \mathcal{R}\big(\mathcal{W}(t)\big)}$ at second order in the time-step $\delta t$. The distance (either $\mathcal{R}$ or $\mathcal{R}_{FS}$) remains small throughout the time-evolution unless a breakdown occurs and can be chosen as a fiducial parameter for a qualitative and quantitative check on the dynamics together with the energy.

The same result can be obtained with a Lagrangian formulation for norm-independent dynamics starting from the following action [36]

$$\mathcal{S} = \int dt \mathcal{L}(\mathcal{W}^*, \mathcal{W}) = \int dt \frac{i}{2}\frac{\langle\dot{\psi}_M^*|\psi_M\rangle - \langle\psi_M^*|\dot{\psi}_M\rangle}{\langle\psi_M^*|\psi_M\rangle} - \frac{\langle\psi_M^*|\hat{H}|\psi_M\rangle}{\langle\psi_M^*|\psi_M\rangle}. \tag{B.3}$$

Stationarity ($\delta\mathcal{S} = 0$) with respect to the variation $\langle\delta\psi_M^*|$ leads to the equation of motion

$$\langle\delta\psi_M^*|\Big(1 - \frac{|\psi_M\rangle\langle\psi_M|}{\langle\psi_M|\psi_M\rangle}\Big)\Big(i\frac{d}{dt}|\psi_M\rangle - \hat{H}|\psi_M\rangle\Big) = 0, \tag{B.4}$$

from which the Euler-Lagrange Eqs. (3) can be derived straightforwardly.

# C  Translation invariance

In this appendix we outline how translation-site invariance is implemented in the RBM wavefunction. For the square lattice we denote the translation operators as $\hat{T}_\xi$, with $\xi = \{x, y\}$.

Given a spin configuration $|S\rangle = |S_1, S_2, \ldots S_N\rangle$, the action of the translation operators can be written as $\hat{T}_\xi |S\rangle = |S'_\xi\rangle$, where $|S'_\xi\rangle$ is the state obtained from $|S\rangle$ after shifting all the spins by one site along the $\xi$ direction of the lattice.

The Heisenberg model and the perturbation Eq. (9) are both invariant under the action of the translation operators $\hat{T}_x$ and $\hat{T}_y$, which means that the total Hamiltonian of the system $\hat{H}_{tot} = \hat{H} + \delta\hat{H}$ satisfies $[\hat{H}_{tot}, \hat{T}_{x,y}] = 0$ (and $[\hat{T}_x, \hat{T}_y] = 0$). Therefore, $\hat{H}_{tot}$, $\hat{T}_x$ and $\hat{T}_y$ admit a common set of eigenstates, denoted with $\{|\psi_{\mathbf{k}}\rangle\}$. The action of the translation operators on such states is given by $\hat{T}_\xi |\psi_{\mathbf{k}}\rangle = \lambda_\xi |\psi_{\mathbf{k}}\rangle$. Since the system under study is finite, and we are employing periodic boundary conditions, we have that

$$\hat{T}_x^L |\psi_{\mathbf{k}}\rangle = |\psi_{\mathbf{k}}\rangle, \quad \hat{T}_y^L |\psi_{\mathbf{k}}\rangle = |\psi_{\mathbf{k}}\rangle. \tag{C.5}$$

This implies that $\lambda_\xi = e^{ik_\xi}$, with $k_\xi = 2\pi m_\xi/L$, $m_\xi = \{-L/2+1, -L/2+2\ldots, L/2\}$. It follows that the Hilbert space divides into $L^2$ different sectors. Within the RBM representation, it is possible to enforce that the RBM wavefunction lives in one of these sectors by imposing

$$\psi_M(\hat{T}_\xi S) = \langle S|\hat{T}_\xi|\psi_M\rangle = \lambda_\xi \psi_M(S). \tag{C.6}$$

For the simulations presented in the main text, the sector $k_x = k_y = 0$ is of particular importance. In this case $\psi_M(S') = \psi_M(S)$ for each state $|S'\rangle$ obtained from a given state $|S\rangle$ by the (repeated) action of the translation operators $\hat{T}_\xi$. This results into a set of conditions on the network parameters. For $\alpha = 1$ ($M = N$) and for $b_j = b$, $a_i = a$, we obtain $\psi_M(S') = \psi_M(S)$ by requiring $\prod_{j=1}^M (\sum_{i=1}^N W_{ij} S'_i) = \prod_{j=1}^M (\sum_{i=1}^N W_{ij} S_i)$. Since there are at most $L^2$ inequivalent $|S'\rangle$ for a given $|S\rangle$, the above condition on $W_{ij}$'s is satisfied by a set of $N$ independent parameters. In our code we take $W_{1j} \equiv W_j$, $j = 1, \ldots, N$ as independent parameters and the other weights $\{W_{2,1} \cdots W_{2,N} \cdots W_{N,1} \cdots W_{N,N}\}$ are defined according to

$$W_{ij} = \hat{T}_y^{Q((i-1)/L)} \hat{T}_x^{i-1} W_j, \tag{C.7}$$

where $Q$ indicates the quotient function and the translation operators act on the index $j$ of $W_j$. For $\alpha > 1$, the procedure is repeated with $W_{1,N+1}, \ldots, W_{1,2N}$ as next set of independent parameters from which $\{W_{2,N+1} \ldots W_{2,2N}, \ldots W_{N,N+1} \ldots W_{N,2N}\}$ are obtained, and so on, with $W_{1,M-N+1}, \ldots, W_{N,M}$ the last set of independent parameters. A different but equivalent approach can be found in [12].

# D  Sample code

Together with this paper we provide in [14] an easy to use implementation of the RBM approach in the Julia language, version 0.6.1 [37]. With the code termed ULTRAFAST it is possible to (i) find the variational ground state energy and wavefunction of the antiferromagnetic Heisenberg model on the square lattice; (ii) time-evolve a given initial state under the perturbation Eq. (9); (iii) evaluate spin correlation functions using the optimized parameters from (i) and (ii). This suffices to reproduce the results shown in the paper and the code can be easily extended to other spin models and different excitation protocols.

To run ULRAFAST, first install Julia following the instructions in [37]. Then install the code by downloading in a suitable working directory the files given in [14]. Julia can be run either from an interactive session Read-Eval-Print Loop ("REPL") or from the command line. To execute ULTRAFAST in the REPL, double-click the Julia executable and type

```
julia> include("run.jl")
```

From the command line, open a terminal and type

```
$ path/to/julia    path/to/run.jl
```

The code features parallel computation [40]. To run ULTRAFAST over N processes on the REPL, type

```
julia> addprocs(N)
julia> include("run.jl")
```

while on the command line, simply type

```
$ path/to/julia  -pN   path/to/run.jl
```

To start a simulation, the neural network and the physical problem to solve need to be initialized. This can be done in the file "model.jl". An example of "model.jl" is given below

```
#Set Neural Network
n_spins = 16              #number of spins (visible units)
α = 4                     #ratio hidden units/visible units
const pbc = true        #pbc=true for periodic boundary conditions, otherwise false
#Set symmetries
#Uncomment the symmetry you want to employ
#Sym = "No symmetry"
Sym = "Translation symmetry"
mag0 = true              #true for sampling from zero magnetization sector, otherwise false
const n_flips = 2        #spin flips in the monte carlo sampling. Set n_flips=2 for mag0=true
```

Here the system size is $N = 4 \times 4$ and $\alpha = 4$. Periodic boundary conditions (pbc) and translation-site symmetry have been selected (pbc = true, Sym = "Translation symmetry"). The zero-magnetization sector is chosen (mag0 = true); in this way only zero-magnetization states are sampled. "n_flips=2" allows for two spin flips in the Monte Carlo sampling.

In the script "run.jl" you can choose to run both the ground state and the dynamic optimization. Ground state optimization starts by calling the function gs_optimization(), which requires the number of Monte Carlo samples, the number of iterations and the learning rate as input. At the end of the optimization the optimal parameters $\mathcal{W}$ are stored in the file "W_rbm_nspins_alpha.jl". Dynamics is run by calling run_dynamics(). Analogous to the ground state optimization, this requires the number of Monte Carlo samples, the total evolution time and the time-step of the numerical time-integration as input. The variable "Init" is an array with the parameters of the initial state wave function. It can be initialized either by using the optimal parameters found in the ground state optimization (default option) or by choosing one of the pre-optimized wavefunction given in [14]. For the latter define: Init = readdlm("W_RBM_nspins_alpha_ti.jl"), where nspins (number of spins) and alpha must be chosen according to "model.jl". The functions GS_obs() and and spincorr_d() allow to measure $\langle \hat{S}_i \cdot \hat{S}_j \rangle$ for any given $i$ and $j$ respectively in the ground state or along the time-evolution. The function GS_obs() also provides the ground state energy per spin.

```
########## GROUND STATE OPTIMIZATION ##################
nsweeps = 1000        #number of monte carlo samples
n_iter = 200          #number of iterations
γ = 0.005             #step size during optimization

gs_optimization(n_iter,nsweeps,gamma)              #run a ground state optimization
writedlm("W_rbm_$(nspins)_(nhv[1]).jl",W_RBM)      #save W_RBM in "W_rbm_nspins_alpha.jl"

nsweeps_gs = 10000              #number of samples for g.s. observables evaluation
m = 1; n = 2;                   #select indices of spin-spin correlation function < S_m S_n >
GS_obs(W_RBM,nsweeps_gs,m,n)    #calculate energy per spin and <S_m S_n > in the g.s.
########## UNITARY DYNAMICS ##################
nsweeps_d = 2000                #number of Monte Carlo sweeps in the time-evolution
length_int = 1.                 #time-length of the time integration
step_size = 0.0025              #time-step of the time integration

Init = W_RBM                    #set the optimized parameters as initial wavefunction
run_dynamics(heun,Init)         #run the time-evolution with initial parameters Init
########## OBSERVABLE EVALUATION ################
nsweeps_obs = 10000             #number of samples for the evaluation of <S_i S_j>
i = 1; j = 2;                   #select indices of spin-spin correlation function <S_i S_j>

spincorr_d = spincorr_d(W_RBM_t,nsweeps_obs,i,j) #evaluate time-evolution of <S_i S_j>
```

For reference we provide numerical data of the ground-state optimization that can be reproduced with the code provided. In Table 1 the variational ground state energies $E(L)$ for different system sizes and $\alpha$ are shown. Table 2 shows the spin-spin correlation functions $\langle \hat{S}_i \cdot \hat{S}_{i+R} \rangle$ for different system sizes.

The code provided can be adopted to efficiently simulate larger system sizes than those studied in the main text. To validate this, in Fig. 5 we show the total time required for a step of a time-dependent optimization for system sizes up to $N = 900$ and $\alpha = \{2, 4\}$. A fixed number of $2 \times 10^4$ samples is used for the optimization, and parallelization is exploited in our cluster machine featuring two AMD EPYC 7601 32-Core processors. Fig. 5 shows that the computational time scales only quadratically with system size; this follows from the fact that for fixed $\alpha$ and number of samples, the more demanding tasks of the optimization, which are the sampling of the energy gradients $\mathcal{F}_k$ and the covariance matrix $S_{kk'}$, depend both quadratically on $N$, if translation invariance is implemented. Studying even larger systems is also feasible by exploiting massively parallel computing, which gives a linear reduction of the computational time with the number of cores exploited. We also stress that the time required for the optimization can be reduced further if other symmetries compatible with the excitation protocol are implemented. An example of this is the two-fold (180°) rotational symmetry which is not broken by the excitation Eq. (9).

Table 1: Ground state energy for different $\alpha$ and different system sizes. Evaluation of the ground state energy is done sampling $10^5$ states. Errors are calculated as statistical errors on the Monte Carlo sampling.

| $N$ | $\alpha = 1$ | $\alpha = 2$ | $\alpha = 4$ | $\alpha = 8$ | $\alpha = 16$ |
|---|---|---|---|---|---|
| 16 | -0.6981(2) | -0.70014(5) | -0.70075(5) | -0.70156(1) | -0.701770(8) |
| 36 | -0.67305(8) | -0.67732(6) | -0.67808(5) | -0.67857(2) | -0.67848(1) |
| 64 | -0.66803(2) | -0.67082(4) | -0.67227(3) | -0.67285(2) | -0.67291(9) |
| 100 | -0.66661(8) | -0.66937(2) | -0.67039(2) | -0.67076(2) | -0.670850(7) |
| 144 | -0.66597(3) | -0.66893(3) | -0.66965(2) | -0.66998(1) | -0.670101(7) |
| 196 | -0.66586(3) | -0.66840(2) | -0.66926(1) | -0.66951(1) | -0.669750(4) |

Table 2: Ground state spin correlation functions $\langle \hat{S}_i \cdot \hat{S}_{i+R} \rangle$ for different system sizes. Evaluation of the correlations is done by sampling $10^6$ states. For $N = 16$ we used $\alpha = 10$, while for larger $N$ we used $\alpha = 16$. Errors are calculated as statistical errors on the Monte Carlo sampling.

| $N$ | $\langle \hat{S}_i \cdot \hat{S}_{i+R} \rangle$ |
|---|---|
| 16 | 0.1798(2) |
| 36 | 0.1528(3) |
| 64 | 0.1386(4) |
| 100 | 0.1289(3) |
| 144 | 0.1238(4) |

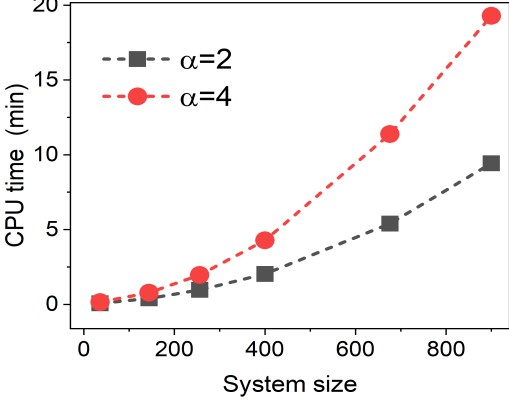

Figure 5: Time required for a single optimization step versus system size. The number of samples for the optimization is kept fixed to $2 \times 10^4$ for each data point. The sizes addressed are: $N = \{36, 144, 256, 400, 676, 900\}$, where $N$ is the number of spins. Simulations are performed in our local cluster machine exploiting 60 cores in parallel during the whole optimization process. The plot clearly shows that for fixed $\alpha$ the scaling is quadratic in the number of spins.

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
