# Peer review of "Investigating ultrafast quantum magnetism with machine learning"

_SciPost Physics, doi:SciPost Phys. 7, 004 (2019)_

## Round 2 · Referee Report · Anonymous (Referee 1) · 2019-4-5

Strengths

1- Timeless investigation of RBM states for strongly-correlated systems.

2- Accurate numerical calculations

3- Well written

Weaknesses

1- Some minor points should be addressed (see report)

Report

The paper is nice and well written. Just a few comments

1) I do not like the fact the nomenclature of ``reinforcement learning'' for the optimization technique. Even though this is a fancy name, the optimisation is just standard, according to the Monte Carlo community (see ref.26).

For the real-time evolution a VMC approach was proposed in Scientific Reports 2, 243 (2012) for a Bose-Hubbard model. I think that this paper should be cited.

2) Are the variational parameters real of complex for the static calculations? Is the Marshall sign imposed?

3) The standard way to define the energy accuracy is to normalize |E_vmc-E_0| by E_0 and not by E_vmc.

Requested changes

1) Do not use ``reinforcement learning'' and add the reference.

2) Specify if W is real or complex for the static calculations.

3) Change the normalization in the accuracy.

---

## Round 2 · Referee Report · Anonymous (Referee 2) · 2019-4-19

Strengths

1- The results presented are new.

2- The technicalities are well-covered in the paper.

Weaknesses

1 - In addition to the energy and the magnetization, the authors could consider other ground state static observables, e.g., magnetic susceptibility, to check the accuracy of the method.

Report

The paper by Fabiani et. al. investigates the accuracy of a recently proposed quantum many-body variational method based on the restricted Boltzmann Machine (RDM) neural network. As point out in the introduction, this method has the potential to efficiently simulate static and dynamic properties of many-body wave functions in any dimension. However, the efficiency of the RDM method in simulations of dynamical properties was not tested in dimensions higher then one. The introduction motivates well this point.

In the paper, the RDM method is applied to study static and some dynamical properties of the prototypical two-dimensional Heisenberg model (HM); the results are then validated with other exact (or approximated) methods.

The results presented by the authors provide relevant information about the the efficiency of the restricted Boltzmann Machine in two-dimensions.

Requested changes

1 - The authors mention that for larger systems, already for $\alpha = 4$ "convergence is reached within Monte Carlo error". Is this a general feature of the method, i.e., larger systems requires smaller $\alpha$ for convergence? or just a numerical observation for this specific case? I think the authors should comment about this on the manuscript.

---

## Round 2 · Referee Report · Anonymous (Referee 3) · 2019-4-27

Strengths

  • the RBM results are benchmarked against both numerical methods (ED, QMC) as well as theory predictions (spin wave theory).
  • the results are new and will contribute to developing the field.
  • the paper is well written and easily accessible to non experts.

Weaknesses

  • the RBM results for dynamics are not as good as one would like them to be (meaning that even at $\alpha=6$ there is a visible mismatch with ED in Fig 2c).

Report

The paper "Investigating ultrafast quantum magnetism with machine learning" by Fabiani and Mentink applies the Restricted Boltzmann Machine variational ansatz for quantum many-body states to study static and dynamic properties of the two-dimensional Heisenberg model on a square lattice: for static properties the authors report agreement with Quantum Monte Carlo results in the thermodynamical limit; for dynamic properties, they "find excellent agreement with exact diagonalization" for small systems, and compare the performance of RBMs to interacting spin-wave theory for larger systems. A major message of the paper is that RBMs allow one to access to much bigger systems than feasible before, which hints at the potential to study quantum many-body dynamics of large spin systems, important for understanding magnetic materials, and strongly-correlated out of equilibrium setups. An open source version of the code termed “ULTRAFAST” is provided by the authors.

The reported results are solid, since the authors benchmarked them against various theory and numerical approaches. I believe the paper constitutes an interesting study which will be of good use to the community.

Requested changes

Suggestions:

  • the authors write: "For dynamics we adopt a time dependent variational scheme where at each time-step the angle between the variational evolved state and the exactly evolved state is minimized.": at this stage the reader may wonder: if no exact simulation is available due to large system sizes, how can this method work? Could the authors elaborate on this?

  • "and the optimization routine becomes equivalent to minimizing the expectation value of the energy for normalized wavefunctions": the word normalized may misleadingly suggest that the normalization of the wavefunciton is required, which is not the case.

  • "In both Eq. (7) and Eq. (8), we retain only first order terms.": I assume order here refers to the expansion in $L^{-1}$? Does this mean that the term proportional to $a$ in Eq(7) is disregarded?

  • "In particular we checked that for α ≤ 10 and αN ≤ 104 [26], system sizes above 30 × 30 spins are feasible in reasonably accessible CPU time on our local cluster nodes. Such system sizes are far beyond the capabilities of exact diagonalization.": could the authors show some of the data they have (maybe in the Appendix) to back up these statements? One thing the authors may want to comment on explicitly is that this is feasible because the RBM ansatz does not have to return properly normalized amplitudes [otherwise there would be an issue with the amplitudes being smaller than machine precision at such Hilbert space sizes].

Misc:

  • "represented by means of a Restricted Boltzmann Machine (RBM), which is a two-layer Artificial Neural Network (ANN)": in the ML community, there is a formal difference between a feed-forward deep neural network and an RBM. For the benefit of the readers, I suggest the authors to make the statement more precise in order to avoid confusion.

  • in paragraph after Eq.(1): "Following [12], the wavefunction of the quantum spin system is identified with the probability amplitude Eq. (1), namely ⟨S|ψM ⟩ ≡ ψM (S) = P(S) ...": the wavefunction (even when real-valued) does not represent a probability distribution, but its square does. The authors should comment on how the ansatz (2) can capture negative [or even complex] probability amplitudes [I guess for this, it is essential that the weights and biases are complex-valued even if the Hamiltonian, and thus its states, are all real].

  • "the set of network parameters Wk = {ai,bi,Wij} is trained via a reinforcement learning algorithm": I am aware that this terminology was first employed in Ref.[12] (presumably due to the presence of a feedback loop), yet I believe it is more appropriate to set the training procedure apart from Reinforcement Learning, which is a whole independent branch of ML per se, the formulation of which is based on Markov decision processes. I only have a few minor suggestions for the authors to consider:

Remarks:

  • some recent work on using RBMs for variational wavefunctions try to learn $\log(\psi)$ instead. This offers an advantage because the state amplitudes for large systems can differ by a few orders of magnitude, and taking the log is expected to mitigate this effect.

Figures:

  • caption to Fig 2: a) "At times t 􏰏 2 the RBM dynamics matches well the dynamics from ED even with α = 2" b) "For large simulation times rapid convergence with α is found."

Could the authors quantify a bit what they mean by "matches well" and "rapid convergence"?

c) could the authors add the system size L to the caption?

  • Fig 4: axis label font needs to be adjusted

Typos:

  • Fig 1, caption: "The extrapolations from the fits for several alpha are shown": alpha --> $\alpha$

  • in caption to Table 2, it it said: "Evaluation of the correlations is done by sampling 10^6 states", but in the code snippet above Table 1, I read "nsweeps obs = 10000 #number of samples for the evaluation of <S i S j>". The number of samples do not match :)

  • "Results dynamics": the title of this Sec sounds somewhat cryptic.

  • "Hence, the dominant contribution of this peak originates from modes with large wave numbers that can be well captured on finite systems.": on --> in

---

## Round 3 · Referee Report · Anonymous (Referee 1) · 2019-6-28

Report

The current version of the paper may be published.

---

## Round 3 · Referee Report · Anonymous (Referee 3) · 2019-6-28

Report

The authors answered to all the points I raised and the quality of the paper improved. I, therefore, recommend publication of the manuscript in its present form.

---

## Round 3 · Author Response

We thank the referees for the careful analysis of the manuscript and the constructive suggestions given. We made all the requested changes as follows.

---

## Round 3 · List of Changes

REFEREE 1 - We removed the nomenclature “Reinforcement learning algorithm” in favor of “Variational Monte Carlo algorithm”. - Added reference Sci. Rep. 2, 243 (2012) as suggested by the referee. - In section 3 we comment on the fact that real valued variational parameters are employed for ground state calculations. This is possible because the model considered is bipartite and therefore it is possible to perform a unitary transformation to a new transformed Hamiltonian which has a ground state that can be parametrized with positive coefficients. This is commonly used for instance in SSE Quantum Monte Carlo algorithms. - As pointed out by the referee, we changed the definition of the relative error in section 3, by normalizing it respect to E_QMC. Also Fig. 1(b) has been changed accordingly.

REFEREE 2 - In section 4 we commented on why for large system sizes (L>4), smaller alphas are needed compared to L=4. In general, larger systems do not require smaller alpha. What we observe is specific to the excitation protocol employed in our study.

REFEREE 3 - Right above Eq. (3), we removed the statement “the angle between the exactly evolved state… is minimized” in order to avoid confusion. What was meant with “exactly evolved” is the infinitesimal time-evolution of the state: (1-i*epsilon H) psi. - Below Eq. (3) we eliminated “for normalized wavefunction” to be more accurate and we added that the minimization is norm-independent. - Below Eq. (8) we changed “only first order terms” to “only the leading order correction”. We mean the leading order correction to the L=infinity value, so for the energy we keep the term proportional to L^-3, while for the spin correlations the term proportional to L^-1. - To back up the feasibility to study large system sizes, we added data for L=16 (256 spins) in Fig.4 and changed the text accordingly. In addition, we further supported this with a new Fig. 5 in Appendix D, clarifying the computational time expected to approach larger systems on our local cluster. To be more conservative, we also substituted “above 30x30 spins” with “up to 30x30 spins” in the conclusion. The referee is correct that the RBM ansatz does not rely on the actual normalization of the wave function; indeed the (time-dependent) variational principle we used, is derived for norm-independent dynamics (see e.g. J. Haegeman, T. J. Osborne, and F. Verstraete, Phys. Rev. B 88, 075133 (2013)). - We gave a more formal definition of the RBM in the introduction to distinguish it from a feed-forward deep neural network. - In section 2, above Eq. (2), we stressed that complex-valued network parameters allow to represent negative or complex probability amplitude. - We removed the nomenclature “reinforcement learning”, see referee 1. - Fig. 2: we added the system size L to the caption and we used more specific terms to describe the comparison with ED. - Fig. 4: added integrated structure factor for L=16 and font size of the axis adjusted. We also split it into two figures to improve readability. - The sample code is not intended to be representative for all simulations presented in the paper. Like the other parameters (n_spins, alpha, step_size, etc), the number of samples can be changed by the user in the code provided to achieve the accuracy desired. - Typos are adjusted. - Changed title section 4 to “Spin dynamics”. - We also thank the referee for pointing out the possibility to optimize $log \psi$ instead of $\psi$ itself. Although we did not suffer from the problem mentioned by the referee about wavefunction amplitudes, we think it could be an interesting idea in light of our future steps.

Miscellaneous - We slightly changed the abstract by explicitly mentioning the largest system size addressed in the paper (N=256). - We added a reference on a recent article: Nat. Comm. volume 10, 1756 (2019)

---

## Editorial Decision

published